# Hematopoietic Prostaglandin D Synthase Is Increased in Mast Cells and Pericytes in Autopsy Myocardial Specimens from Patients with Duchenne Muscular Dystrophy

**DOI:** 10.3390/ijms25031846

**Published:** 2024-02-03

**Authors:** Kengo Hamamura, Yuya Yoshida, Kosuke Oyama, Junhao Li, Shimpei Kawano, Kimiko Inoue, Keiko Toyooka, Misaki Yamadera, Naoya Matsunaga, Tsuyoshi Matsumura, Kosuke Aritake

**Affiliations:** 1Laboratory of Chemical Pharmacology, Faculty of Pharmaceutical Sciences, Daiichi University of Pharmacy, Fukuoka 815-8511, Japan; hamamura@phar.kyushu-u.ac.jp; 2Department of Clinical Pharmacokinetics, Faculty of Pharmaceutical Sciences, Kyushu University, 3-1-1 Maidashi Higashi-ku, Fukuoka 812-8582, Japan; yoshida@phar.kyushu-u.ac.jp (Y.Y.); li.junhao.587@s.kyushu-u.ac.jp (J.L.); kawano.shimpei.827@s.kyushu-u.ac.jp (S.K.); matunaga@phar.kyushu-u.ac.jp (N.M.); 3Department of Pharmaceutics, Faculty of Pharmaceutical Sciences, Kyushu University, 3-1-1 Maidashi Higashi-ku, Fukuoka 812-8582, Japan; oyama@phar.kyushu-u.ac.jp; 4Department of Neurology and Rehabilitation Medicine, National Hospital Organization Osaka Toneyama Medical Center, Toneyama 5-1-1, Toyonaka 560-8552, Japan; inoue.kimiko.gs@mail.hosp.go.jp; 5Department of Neurology, National Hospital Organization Osaka Toneyama Medical Center, Toneyama 5-1-1, Toyonaka 560-8552, Japan; toyooka.keiko.yq@mail.hosp.go.jp (K.T.); matsumura.tsuyoshi.kq@mail.hosp.go.jp (T.M.); 6Department of Clinical Research, National Hospital Organization Osaka Toneyama Medical Center, Toneyama 5-1-1, Toyonaka 560-8552, Japan; yamadera.misaki.gj@mail.hosp.go.jp

**Keywords:** Duchenne muscular dystrophy, hematopoietic prostaglandin D synthase, prostaglandin D_2_, mast cell, pericyte, myeloid cell

## Abstract

The leading cause of death for patients with Duchenne muscular dystrophy (DMD), a progressive muscle disease, is heart failure. Prostaglandin (PG) D_2_, a physiologically active fatty acid, is synthesized from the precursor PGH_2_ by hematopoietic prostaglandin D synthase (HPGDS). Using a DMD animal model (*mdx* mice), we previously found that HPGDS expression is increased not only in injured muscle but also in the heart. Moreover, HPGDS inhibitors can slow the progression of muscle injury and cardiomyopathy. However, the location of HPGDS in the heart is still unknown. Thus, this study investigated HPGDS expression in autopsy myocardial samples from DMD patients. We confirmed the presence of fibrosis, a characteristic phenotype of DMD, in the autopsy myocardial sections. Additionally, HPGDS was expressed in mast cells, pericytes, and myeloid cells of the myocardial specimens but not in the myocardium. Compared with the non-DMD group, the DMD group showed increased HPGDS expression in mast cells and pericytes. Our findings confirm the possibility of using HPGDS inhibitor therapy to suppress PGD_2_ production to treat skeletal muscle disorders and cardiomyopathy. It thus provides significant insights for developing therapeutic drugs for DMD.

## 1. Introduction

Duchenne muscular dystrophy (DMD) is a severe X-linked recessive disorder arising from mutations in the dystrophin gene that result in a complete absence of dystrophin protein expression [1]. The global incidence of DMD is estimated at 1 in 5000 boys, establishing it as one of the most prevalent recessive diseases in the world. Women are mostly carriers, and affected men are initially normal at birth; however, progressive and persistent muscle weakness and atrophy develop in the proximal muscles of the extremities in early childhood, gradually extending to distal muscles [2]. Typically, clinical diagnosis is performed within the first few years of life [3,4]. Early manifestations, including a waddling gait, frequent falls, difficulty rising from a sitting position (Gower’s sign), and challenges in climbing stairs, become apparent between the ages of 2 and 3 years [5]. Children affected by DMD undergo delays in gross motor development, with the majority requiring a wheelchair by the age of 10–12 years [6]. Progressive muscle weakness contributes to the development of scoliosis and joint contractures, fostering restrictive lung disease [7]. Thus, assisted ventilation becomes necessary by around 15–20 years of age. However, despite recent advancements in ventilator therapy and optimal care, the majority of DMD patients succumb to heart/respiratory failure between the ages of 20 and 30 [2,8].

Prostaglandin (PG) D_2_ is a bioactive fatty acid that plays a role in sleep regulation in the central nervous system and in inflammation and allergic reactions throughout the body [9,10]. PGD_2_ is generated through the isomerization of PGH_2_ by PGD synthase_,_ whereas PGH_2_ is biosynthesized from arachidonic acid present in phospholipids, which are constituent lipids of biological membranes. The catalytic action of cyclooxygenase (COX) serves as the substrate for this process.

Hematopoietic PGD synthase (HPGDS), one of the PGD_2_ synthases, is distributed in the cytoplasm of mast cells and Th2 cells and catalyzes PGD_2_ production during inflammation and allergic reactions [11,12]. HPGDS belongs to the gene family of σ-type glutathione S-transferase (GST). Like other GSTs, it exists as a homodimer in vivo with a molecular weight of approximately 52 kDa. The enzyme at its core coordinates with Mg^2+^ and indispensably relies on the coenzyme glutathione for the enzymatic reaction, i.e., isomerization of PGH_2_ to PGD_2_ [13]. HPGDS and PGD_2_ have been reported to be highly expressed at injured sites of the skeletal muscles of patients with DMD and model animals, resulting in secondary skeletal muscle damage due to inflammation [14,15,16]. A study using *mdx* mice, a DMD model, has shown that administration of HPGDS inhibitors suppresses the progression of muscle injury [17]. In our previous study, we observed an elevated expression of HPGDS and PGD_2_ in the myocardium during the progression of cardiomyopathy in *mdx* mice, suggesting the potential involvement of PGD_2_ in myocardial damage, extending beyond its role in skeletal muscle [18]. Additionally, our research indicated that a compound designed to specifically enhance the degradation of HPGDS may hold therapeutic promise for addressing both skeletal and cardiac muscle disorders [18]. However, the localization of HPGDS in the heart of DMD patients and its role remain unclear.

Thus, the present study aimed to investigate the localization and potential role of HPGDS in myocardial tissues of patients with DMD. We analyzed the specificity of HPGDS by examining the cells expressing this compound in autopsy myocardial specimens.

## 2. Results

### 2.1. Fibrotic Progression in DMD Autopsy Myocardial Sections

The myocardium, responsible for pumping blood throughout the body, experiences significant mechanical stress. This stress is particularly heightened in patients with DMD. The gradual instability in the cell membranes of the myocardium eventually results in the death of myocytes and the development of fibrosis [19,20].

Consequently, we initially compared the fibrotic areas in autopsy myocardial sections obtained from DMD and non-DMD patients using Masson’s trichrome staining (Figure 1 and Appendix A). We found that significantly more areas were stained in blue (indicating fibrosis) in the DMD group than in the non-DMD group. As shown by the Masson’s trichrome-stained images, fibrosis mainly developed in the myocardium in the DMD group, whereas fibrosis was observed in nonmyocardial cells (e.g., around blood vessels) in the non-DMD group. Thus, we confirmed the presence of fibrosis, a characteristic phenotype of DMD, in the autopsy myocardial sections obtained from DMD patients.

### 2.2. Comparison of HPGDS Levels in DMD Autopsy Myocardial Specimens

Next, we compared the expression levels of HPGDS in the specimens. The mRNA and protein levels of HPGDS in DMD autopsy myocardial specimens were quantified using real-time PCR and Western blotting, respectively (Figure 2). The results showed that the DMD group exhibited higher mRNA and protein levels than the non-DMD group, but this difference was not significant (Figure 2). Additionally, the expression levels of COX1 and COX2, which are located upstream of HPGDS and play a role in the arachidonic acid cascade, were also increased. However, this increase did not reach statistical significance (Appendix A). Similarly, comparing protein expression levels, we found that the expressions of membrane-bound prostaglandin E synthase (mPGES1), COX1, COX2, tryptases, and CD31 were not significantly different among the two groups (Appendix A). Therefore, we analyzed the localization of HPGDS using DMD samples with the highest protein levels.

### 2.3. Localization of HPGDS in Mast Cells, Pericytes, and Myeloid Cells of the Heart

HPGDS and PGD_2_ are highly expressed at injury sites of the skeletal muscles of DMD patients and model animals, contributing to secondary skeletal muscle damage due to inflammation [14,15,16]. Initially, we hypothesized that HPGDS is localized in the myocardium and thus stained the tissues with the myocardial marker protein troponin T. However, HPGDS did not colocalize with troponin T, indicating its absence in the myocardium (bottom left image in Figure 3).

Subsequently, we attempted to identify cells expressing HPGDS by examining overlaid brightfield and HPGDS images. HPGDS was stained red, while cells expressing tryptase (mast cell marker), NG2 (pericyte marker), and CD11b (myeloid cell marker, including monocytes and macrophages) were stained green, revealing colocalization in yellow (Figure 3, upper panel). Notably, HPGDS did not colocalize with α-SMA (vascular smooth muscle marker) or CD31 (vascular endothelial cell marker) (Figure 3, bottom middle, right) expressing cells. These results confirm that HPGDS is localized in mast cells, pericytes, and myeloid cells in the heart of DMD patients.

### 2.4. Increased HPGDS in Mast Cells and Pericytes in DMD Autopsy Myocardial Specimens

Flow cytometry was utilized to evaluate the percentage of HPGDS-positive cells within their respective expressing cell populations. The number of tryptase-positive mast cells were increased in the autopsy myocardial specimens of the DMD group compared with the non-DMD group, although the difference was not significant (Figure 4a). The percentage of CD11b-positive myeloid cells remained unchanged between the two groups (Figure 4c). However, NG2-positive pericytes were significantly increased in the autopsy myocardial specimens of the DMD group compared with the non-DMD group (Figure 4b), indicating an elevated presence of pericytes in the heart of DMD patients.

Upon analyzing the percentage of HPGDS-positive cells, we observed a significant increase in the percentages of Tryptase+/HPGDS+ and NG2+/HPGDS+ in the autopsy myocardial specimens from DMD patients (Figure 4d,e). Conversely, the CD11b+/HPGDS+ ratio exhibited no significant change between the two groups (Figure 4f). Based on these findings, we conclude that HPGDS is heightened in the mast cells and pericytes of myocardial tissues of patients with DMD.

## 3. Discussion

This study revealed that HPGDS levels were elevated in mast cells and pericytes of myocardial specimens obtained from individuals with DMD at autopsy. While previous studies have demonstrated the localization of HPGDS in mast cells [12], our study is the first to reveal the presence of HPGDS in pericytes and to demonstrate an increase in both pericyte proportion and HPGDS expression levels in the heart of patients with DMD. One of the significances of our findings includes the distinct production and action sites of PGD_2_ in the heart compared with the skeletal muscle.

In skeletal muscles, HPGDS and PGD_2_ are highly expressed at the site of injury, leading to secondary skeletal muscle damage, which is resolved within the skeletal muscle itself [14,15,16]. However, our results suggest a different scenario in the myocardium. PGD_2_ production is proposed to occur in mast cells and pericytes, with highly expressed HPGDS around the myocardium or in myeloid cells. Subsequently, PGD_2_ migrates to the myocardium, where it becomes involved in myocardial damage, with fibrosis as the primary phenotype [18]. These findings underscore the potential involvement of PGD_2_, synthesized by specific cell types, in the progression of myocardial damage in DMD. The migration of PGD_2_ from mast cells, pericytes, or myeloid cells to the myocardium adds a layer of complexity to understanding the role of PGD_2_ in cardiac manifestations during DMD, particularly in the context of fibrosis in the myocardium.

PGD_2_ has a short half-life in the body, and its half-life in the blood is less than 1 min, making it extremely difficult to measure. It is usually measured from urinary levels of tetranor-PGDM (tetranor-prostaglandin D metabolite), a metabolite of PGD_2_. Tetranor-PGDM is a useful marker for identifying the progression of systemic pathological inflammation in DMD [15,16]; however, it cannot be measured in autopsy myocardial specimens.

Downstream of the PGD_2_ pathway, PGD_2_ interacts with two G protein-coupled receptors, DP1 and DP2/CRTH2 (chemoattractant receptor homolog expressed on helper type 2 cells)/GPR44, to exert various biological effects [21]. The DP1 and DP2 receptor is found in many tissues but primarily in the gastrointestinal tract, bone marrow, and immune system [22]. Under normal conditions, the heart has a low number of DP1 and DP2 receptors [23]. However, in our previous study, *Dp1* and *Dp2* mRNA levels were found to be significantly upregulated in the hearts of *mdx* mice treated with thyroid hormone (T3) to induce dilated cardiomyopathy [18]. The above results indicate that PGD_2_ produced by HPGDS contributes to cardiac function, especially fibrosis, through DP1 and DP2 receptors. Analysis of the sites of DP1 and DP2 receptor expressions in the heart of DMD patients has not been demonstrated to date, owing to poor antibody performance, but we are currently trying to validate this using T3-treated *mdx* mice.

The comparison of fibrotic areas in non-DMD and DMD tissues using Masson’s trichrome staining confirmed the presence of cardiac fibrosis, a crucial phenotype in DMD pathology, as illustrated in Figure 1. However, it is noteworthy that the autopsy myocardial specimens of the non-DMD group also displayed fibrosis against underlying pathological conditions, including amyotrophic lateral sclerosis, neurofibrillary dementia, distal myopathy, spinal muscular atrophy, anti-SRP antibody positive reaction, and necrotizing myositis (Table 1). Regardless, the percentage of fibrosis was less than 10% in all samples, indicating the absence of significant cardiac fibrosis in the control group (Figure 1 and Appendix A); however, the fibrosis rate found in this study was higher than the previously reported rate of normal samples [24]. This may be because the age at death was inevitably higher in the non-DMD group than in the DMD patient group in the present study (a limitation of this study), indicating an effect of aging [24]. In addition, Masson’s trichrome staining revealed distinct blue areas within the myocardium of the DMD group. In contrast, fibrosis advanced in cells outside the myocardium (e.g., around blood vessels) in the non-DMD group. This suggests that the progression of perivascular fibrosis with age may contribute to the fibrosis area in the non-DMD group. These findings reinforce the integrity of the control group, establishing a baseline for interpreting fibrotic changes in the DMD group.

The comparison of mRNA levels in autopsy myocardial specimens obtained from non-DMD and DMD patients indicated that the expression levels of *COX1* and *COX2* were increased in DMD patients, but this difference was not statistically significant (Appendix A). Similarly, the protein levels of membrane-bound prostaglandin E synthase (mPGES1), COX1, COX2, tryptases, and CD31 did not significantly differ among the two groups (Appendix A). In contrast, *Cox1* and *Cox2* mRNAs have been found to be upregulated in the hearts of T3-treated *mdx* mice [18]. The myocardial specimens in this study might have been influenced by the time elapsed between death and autopsy (Table 1) and potential decomposition, as suggested in past reports [25]. Indeed, in autopsy myocardial specimens from DMD patients, a weak inverse correlation was confirmed between the protein expression level of HPGDS and the duration between postmortem and dissection (exponential approximation, R^2^ = 0.318). In any case, the overall evaluation of autopsy myocardial specimens did not reveal a significant difference between the two groups, prompting a more detailed investigation.

Pericytes, versatile mesenchymal cells located in microvessel walls, have garnered substantial interest as potential therapeutic agents, particularly in the heart, owing to their mesenchymal properties that may contribute to cell repair and regeneration [26]. Consequently, previous studies on cell transplantation therapy for muscular dystrophy have investigated the potential mitigating effect of pericyte-derived cells on muscle damage [27,28]. NG2-positive pericytes were present in significantly higher proportion in autopsy myocardial specimens from the DMD group compared with the non-DMD group, indicating an augmented presence of pericytes in the heart of DMD patients. This elevation is likely associated with fibrosis. Although fibroblasts are generally considered the primary source of myofibroblasts, recent studies have suggested alternative cellular origins [29]. The concept of a pericyte-to-myofibroblast transition has been proposed as a contributing factor in various fibrotic conditions [30,31]. It has been observed that pericytes migrate to injury sites and express profibrotic genes, a phenomenon consistent with increased vascular leakage after myocardial infarction [32]. In essence, there appears to be an association between fibrosis and pericytes, and it is plausible that an increased number of pericytes in the myocardium of DMD patients is linked to the higher number of fibrotic sites. However, the physiological significance of why PGD_2_ production is increased in pericytes and mast cells in the heart of DMD individuals is still unknown and warrants further research.

In conclusion, this study demonstrated that HPGDS is increased in mast cells and pericytes of myocardial specimens obtained from individuals with DMD at autopsy. Currently, domestic phase III clinical trials of HPGDS inhibitors are underway to assess the efficacy and safety of these inhibitors for DMD pathologies, using the inhibition of limb muscle weakness and atrophy in DMD patients as indicators (NCT04587908/jRCT2041200055) [33,34]. The findings of the present study provide evidence that PGD_2_ production suppression therapy may be effective not only for skeletal muscle disorders but also for cardiomyopathy. This information serves as a crucial foundation for drug development, offering insights into the myocardial targets of HPGDS, which include mast cells, pericytes, and myeloid cells. The successful development of drugs based on these results could represent a significant advancement for patients with muscular dystrophy, particularly those for whom heart failure is the primary cause of death. Moreover, it may hold promise for patients with other cardiomyopathies, providing additional treatment options.

## 4. Materials and Methods

### 4.1. Autopsy Myocardial Specimens

This study was conducted in accordance with the Declaration of Helsinki and approved by the Institutional Review Board of National Hospital Organization Osaka Toneyama Medical Center (approval no. TNH-R-2021006; approved on 19 April 2021) and the Institutional Ethics Committee of Daiichi University of Pharmacy (approval no. R03-0001). We used 10 autopsy myocardial specimens, including 5 from DMD cases and 5 from non-DMD cases, deposited in the Psychiatric and Neurological Disease Brain Bank at Osaka Toneyama Medical Center (Table 1).

### 4.2. Quantitative Reverse Transcription (RT)-Polymerase Chain Reaction (PCR)

Total RNA from the specimens was extracted using ISOGEN (Nippon Gene Co., Ltd., Tokyo, Japan), and homogenization was performed using a high-speed bead cell disruption system (Bertin Technologies, Montigny-le-Bretonneux, France). cDNA was synthesized with the ReverTra Ace qPCR RT kit (Toyobo, Osaka, Japan) and amplified by PCR. The RT-PCR analysis was performed on diluted cDNA samples using the FastStart Essential DNA Probes Master (Nippon Genetics Co., Ltd., Tokyo, Japan) with a LightCycler 96 System (Roche, Basel, Switzerland). Data were normalized to the expression levels of the control GAPDH mRNA. Appendix A shows the list of primer sequences used in this study.

### 4.3. Western Blotting

The same samples used for mRNA extraction were used for protein isolation. Proteins were extracted from the intermediate and organic layers after chloroform precipitation using ISOGEN, following the manufacturer’s instructions. Protein samples after heat denaturation were separated using sodium dodecyl sulfate-polyacrylamide gel electrophoresis (Gel Factory mini; DRC. Co., Ltd., Tokyo, Japan), followed by transferring to polyvinylidene difluoride membranes (Immobilon-P; Merck Millipore, Burlington, MA, USA). The transfer membranes were blocked by immersing in Blocking One (Nacalai Tesque, Kyoto, Japan) for 30 min and then subjected to primary antibody reaction with anti-HPGDS antibody (160013; Cayman Chemicals, MI, USA) [35,36] or anti-GAPDH antibody (016-25523; FUJIFILM Wako Pure Chemical Corporation, Osaka, Japan) at a dilution ratio of 1:10,000 in Solution I (Toyobo, Osaka, Japan) overnight. Secondary antibody reactions were conducted with goat anti-rabbit IgG antibody (ab205718; Abcam, Cambridge, UK) in Solution II (Toyobo) at a dilution of 1:5000 for anti-HPGDS antibody and with anti-mouse HRP-labeled secondary antibody (NA931; GE Healthcare, Chicago, IL, USA) in Solution II at a dilution of 1:10,000 for anti-GAPDH antibody for 1 h at room temperature (15–25 °C). Chemiluminescence was performed using Chemi-Lumi One reagent (Nacalai Tesque), and detection was carried out using a ChemiDocTM XRS Plus imaging system (Bio-Rad Laboratories, Inc., Hercules, CA, USA).

### 4.4. Preparation of Paraffin Sections and Deparaffinization

Autopsy myocardial specimens were placed in a Uni-cassette (Sakura Finetech Japan Co., Ltd., Tokyo, Japan) and treated with 10% neutral buffered formalin solution (pH 7.4, 062-01661: Fujifilm Wako Pure Chemical Industries, Ltd., Osaka, Japan). They were soaked overnight at room temperature and then embedded in paraffin. Sections (7 μm thick) were cut using a microtome (REM-710: Yamato Kohki Industrial Co., Ltd., Saitama, Japan), which were then mounted on APS-coated slides and deparaffinized before use. Images were acquired using a Keyence BZ-810 fluorescence microscope (Keyence, Osaka, Japan).

### 4.5. Masson Trichrome Staining

Masson trichrome staining was performed following the protocol outlined in the Modified Masson’s trichrome staining kit (ScyTek Laboratories, Inc., Logan, UT, USA).

### 4.6. Fluorescent Immunostaining

After deparaffinization, the sections were antigen-activated with Histo VT One (Nacalai Tesque) and blocked using Blocking One Histo (Nacalai Tesque). Primary antibody reactions were performed with anti-HPGDS (Cayman Chemicals, Ann Arbor, MI, USA) at a dilution ratio of 1:2000 in Solution A (Toyobo) and anti-Tryptase (ab2378: Abcam), anti-NG2 (sc-80003: Santa Cruz Biotechnology, Inc., Dallas, TX, USA), anti-CD11b (66519-1-Ig: Proteintech Group, Inc., Rosemont, IL, USA), anti-Troponin T (MAB18742: Bio-Techne Corp., Minneapolis, MN, USA), anti-α-SMA (sc-53142: Santa Cruz Biotechnology, Inc.), and anti-CD31 (sc-376764: Santa Cruz Biotechnology, Inc.) at a dilution ratio of 1:500 in Solution A overnight at 4 °C. Secondary antibodies were reacted with goat anti-rabbit IgG H&L (Alexa Fluor^®^ 647) (ab150083: Abcam) for anti-HPGDS and goat anti-mouse IgG H&L (Alexa Fluor^®^ 488) (ab150113: Abcam) for others in Solution A (Toyobo) (dilution 1:1200) for 1 h at room temperature. Sections were mounted with Vectashield Hard Set Mounting Medium with DAPI (H-1500: Vector Laboratories, Inc., Newark, CA, USA). To confirm the specificity of the antibodies, we also verified that no fluorescence was observed in the negative control without the primary antibody.

### 4.7. Flow Cytometry

Autopsy myocardial specimens were prepared for flow cytometry using the Minute™ Cell Suspension Isolation Kit for Fresh/Frozen Tissues (CS-031: Invent Biotechnologies, Inc., Plymouth, MN, USA). Subsequently, FACS buffer (5% BSA and 0.01% sodium azide in PBS) provided in the kit was used as the solvent. Isolated cells were fixed with 4% paraformaldehyde, permeabilized with methanol, and treated with True-Stain Monocyte Blocker (426102: BioLegend, San Diego, CA, USA), followed by the same antibodies used in the immunofluorescent sections. Primary antibodies were diluted 100-fold (only anti-HPGDS was diluted 500-fold), and secondary antibodies were diluted 500-fold. Stained cells were subjected to flow cytometry (CytoFLEX: Beckman Coulter, Inc., Tokyo, Japan), and the results were analyzed using Kaluza (Beckman Coulter, Inc.). Gating strategies are shown in Appendix A. Gates were set using the results obtained under conditions excluding each antibody.

### 4.8. Statistical Analysis

Results are presented as a mean ± standard error of the mean (S.E.). All statistical analyses were performed using the JMP Pro 17 software (SAS Institute Japan, Tokyo, Japan). Significant differences between two groups was analyzed by unpaired Student’s *t*-test; *p* values < 0.05 were considered significant.

## Figures and Tables

**Figure 1 ijms-25-01846-f001:**
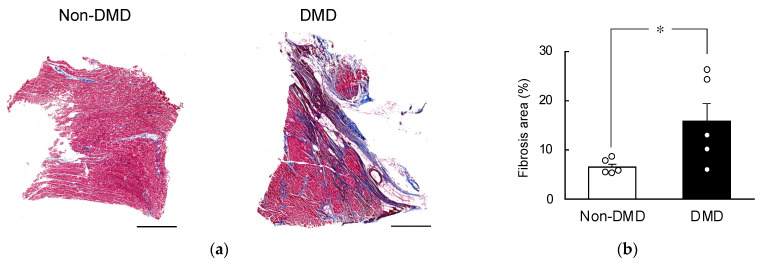
Comparison of fibrosis area using Masson’s trichrome stain. (**a**) Representative Masson’s trichrome stain images of autopsy myocardial specimens obtained from non-Duchenne muscular dystrophy (DMD) and DMD patients. The scale bar indicates 1000 μm. (**b**) Comparison of the percentages of fibrotic areas in non-DMD and DMD tissue sections. Images used for the analysis are shown in Appendix A. Each value is the mean ± S.E. (*n* = 5). * *p* < 0.05 represents significant differences between the two groups (unpaired Student’s *t*-test).

**Figure 2 ijms-25-01846-f002:**
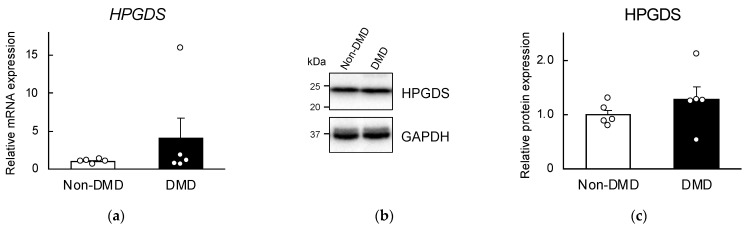
Comparison of hematopoietic prostaglandin D synthase (HPGDS) mRNA (**a**) and protein (**b**,**c**) levels of autopsy myocardial specimens obtained from non-DMD and DMD patients. (**b**) Representative Western blotting images of HPGDS and GAPDH proteins. Uncropped images are available in Appendix A. In (**a**,**c**), each value represents the mean ± S.E. (*n* = 5). There were no significant differences among the two groups (unpaired Student’s *t*-test).

**Figure 3 ijms-25-01846-f003:**
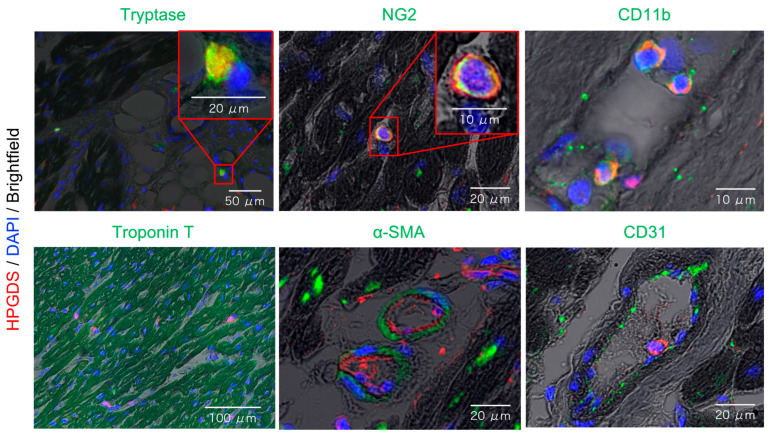
Representative images of fluorescent immunostaining to identify HPGDS-expressing cells of autopsy myocardial specimens obtained from DMD patients. All images have the same settings for HPGDS (red), DAPI (blue), and brightfield, along with their respective green settings. The original images without superimposition are available in Appendix A. The scale bars are indicated on each image.

**Figure 4 ijms-25-01846-f004:**
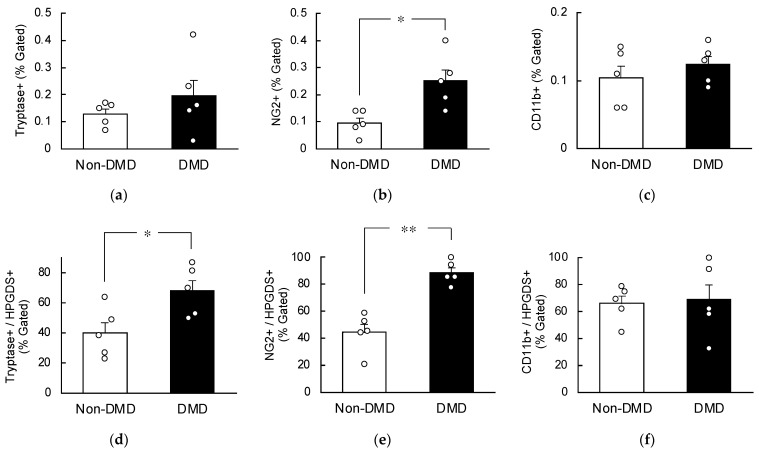
Comparison of the percentage of Tryptase- (**a**), NG2- (**b**), CD11b- (**c**), Tryptase/HPGDS- (**d**), NG2/HPGDS- (**e**), and CD11b/HPGDS-positive cells (**f**) in HPGDS-expressing cells of the autopsy myocardial specimens obtained from non-DMD and DMD patients. Gating strategies for flow cytometry are shown in Appendix A. In all panels, each value is the mean ± S.E. (*n* = 5). ** *p* < 0.01, * *p* < 0.05 indicates significant differences among the two groups (unpaired Student’s *t*-test).

**Table 1 ijms-25-01846-t001:** Patient characteristics.

Non-DMD					
Number	Age	Gender	Postmortem Time at Autopsy	Diagnosis	Gene Search
#1	53	M	2 h 24 min	Amyotrophic lateral sclerosis	-
#2	79	F	1 h 20 min	Neurofibrillary dementia	-
#3	66	M	14 h 22 min	Distal myopathy	-
#4	61	M	4 h 10 min	Spinal muscular atrophy	SMN1 exon 7 deletion
#5	61	M	18 h 00 min	Anti-SRP antibody positivenecrotizing myositis	-
**DMD**					
**Number**	**Age**	**Gender**	**Postmortem Time at Autopsy**	**Diagnosis**	**Gene Search**
#1	29	M	12 h 01 min	DMD	exon 5–16 deletion (in-frame)
#2	28	M	17 h 07 min	DMD	exon 52-deletion
#3	41	M	1 h 33 min	DMD	No data
#4	25	M	3 h 26 min	DMD	exon 52-deletion
#5	39	M	2 h 24 min	DMD	Deletion/duplication (−)

## Data Availability

Data from this study are available from the corresponding author upon reasonable request.

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
