# Peer review of "Hematopoietic Prostaglandin D Synthase Is Increased in Mast Cells and Pericytes in Autopsy Myocardial Specimens from Patients with Duchenne Muscular Dystrophy"

_ijms, 2024, doi:10.3390/ijms25031846_

Round 1
Reviewer 1 Report
Comments and Suggestions for Authors
The first cycle of revision for the manuscript titled Hematopoietic Prostaglandin D Synthase is Increased in Mast Cells and Pericytes in Autopsy Myocardial Specimens from Patients with Duchenne Muscular Dystrophy by Hamamura et al.
In this manuscript, the authors elegantly measured the HPGSD in myocardial biopsies collected post-mortem in patients affected by Duchenne Muscular Dystrophy. The author found that HPGSD is mainly expressed in infiltrating immune cells rather than the myocardium, as seen in the mdx animal model. It is indeed still debated if chronic levels of inflammation due to the degeneration of muscle fibers are the culprit to myocardial damage instead of a direct effect of the genetic disease of the heart itself. To this point, the authors have mentioned in the manuscript a paper from their group in which they measured HPGDS levels in the myocardium of mdx mice. Unfortunately, there is no reference in the manuscript, and I could not find it in the literature. Please provide the reference at the end of lines 70-71 on page 2.
While the methodology is accurately described, it appears to this reviewer that the description of the patients (especially the controls) is very superficial. Several questions remain unanswered: The authors should provide information regarding the populations chosen. Why I understand all the difficulties of recruiting human cardiac specimens. More information regarding the age, sex, and cause of death of this patient should be provided, especially considering that the author reported finding fibrosis around 10% despite underlying pathological conditions; whether this is related to the heart or not remains unclear.
Author Response
Reviewer #1
The first cycle of revision for the manuscript titled Hematopoietic Prostaglandin D Synthase is Increased in Mast Cells and Pericytes in Autopsy Myocardial Specimens from Patients with Duchenne Muscular Dystrophy by Hamamura et al.
We express our sincere gratitude for your thoughtful suggestions and insightful comments. In response to your valuable feedback, we have thoroughly revised the manuscript now. We have provided detailed point-by-point responses below.
In this manuscript, the authors elegantly measured the HPGSD in myocardial biopsies collected post-mortem in patients affected by Duchenne Muscular Dystrophy. The author found that HPGSD is mainly expressed in infiltrating immune cells rather than the myocardium, as seen in the mdx animal model. It is indeed still debated if chronic levels of inflammation due to the degeneration of muscle fibers are the culprit to myocardial damage instead of a direct effect of the genetic disease of the heart itself. To this point, the authors have mentioned in the manuscript a paper from their group in which they measured HPGDS levels in the myocardium of mdx mice. Unfortunately, there is no reference in the manuscript, and I could not find it in the literature. Please provide the reference at the end of lines 70-71 on page 2.
As you rightly point out, there is an ongoing debate regarding whether chronic inflammation resulting from muscle fiber degeneration is the primary cause of myocardial damage. Nevertheless, it is well-established that inflammation promotes fibrosis in the myocardium of DMD patients [19,20]. Furthermore, previous research has indicated that mast cells are increased in cardiomyopathy (Levick SP, et al., Cardiovasc Res. 2011 Jan 1;89(1):12-9.) and PGD2 is involved in mast cell activation (Zhang S, et al., Am J Physiol Gastrointest Liver Physiol. 2013 May 15;304(10):G908-16.). Thus, we believe that HPGDS can serve as a potential therapeutic target.
I would also like to express gratitude for bringing to my attention the omission of cited references. I have now included the reference regarding our previous study on measuring HPGDS levels in the myocardium of mdx mice (page 2, line 73).
While the methodology is accurately described, it appears to this reviewer that the description of the patients (especially the controls) is very superficial. Several questions remain unanswered: The authors should provide information regarding the populations chosen. Why I understand all the difficulties of recruiting human cardiac specimens. More information regarding the age, sex, and cause of death of this patient should be provided, especially considering that the author reported finding fibrosis around 10% despite underlying pathological conditions; whether this is related to the heart or not remains unclear.
We sincerely apologize for any oversight in addressing our readers’ concerns. Information such as age and gender at the time of death has been presented in Table 1, with careful measures taken to ensure the confidentiality of personal details. The non-DMD group comprised patients with underlying diseases other than heart disease, selected from those registered in the Psychiatric and Neurological Disease Brain Bank at Osaka Toneyama Medical Center.
In the previous version of the manuscript, Table 1 was placed in the discussion section adhering to the journal's guidelines of inserting graphics after the paragraph in which they are first cited. However, for enhanced accuracy in conveying information, Table 1 has been relocated to “4.1. Autopsy Myocardial Specimens” in the Materials and Methods section (pages 7–8, lines 278–281).
Regarding the observation of approximately 10% fibrosis rate in the non-DMD group, please refer to the images in Figures 1 and S1 and Table 1. We are considering three possible causes for this observation:
(i) The age at death in the non-DMD group was inevitably higher than that in the DMD group, which is a limitation of this study. In the MT-stained images of the DMD group, blue areas are clearly visible in the myocardium. However, in the non-DMD group, fibrosis progressed in cells that are not part of the myocardium (around blood vessels). We attribute this observation to aging (Shirakabe A, et al., Circ Res. 2016 May 13;118(10):1563-76.). Therefore, we believe that fibrosis around blood vessels may have contributed to the slightly higher fibrosis rate observed in non-DMD patients in this study.
(ii) The blue threshold setting during the image analysis stage may also be a contributing factor.
(iii) The collection location of autopsy myocardial specimens may vary depending on the individual.
Based on your suggestion, we have added the following statement to the Discussion section.
“Regardless, the percentage of fibrosis was less than 10% in all samples, indicating the absence of significant cardiac fibrosis in the control group (Figure 1, Figure S1); however, the fibrosis rate found in this study was higher than previously reported rate of normal samples [24]. This may be because the age at death was inevitably higher in the non-DMD group than in the DMD patient group in the present study (a limitation of this study), indicating an effect of aging [24]. In addition, Masson's trichrome staining revealed distinct blue areas within the myocardium of the DMD group. In contrast, fibrosis advanced in cells outside the myocardium (e.g., around blood vessels) in the non-DMD group. This suggests that the progression of perivascular fibrosis with age may contribute to the fibrosis area in the non-DMD group. These findings reinforce the integrity of the control group, establishing a baseline for interpreting fibrotic changes in the DMD group.” (Page 6, lines 209–220).

Reviewer 2 Report
Comments and Suggestions for Authors
Author present an interesting article that argues for involvement of HPGDS in the progression of DMD. The results are fairly interesting; however, I have three major issues.
1. I cannot find supplementary files.
2. Discussion is heavily used to present additional Results. Results should be presented in the Results section.
3. Too many instances of (unpublished data). If the data is unpublished, it is good time to publish that data. If it is not possible, related claims need to be withdrawn.
Author Response
Reviewer #2
Author present an interesting article that argues for involvement of HPGDS in the progression of DMD. The results are fairly interesting; however, I have three major issues.
We sincerely appreciate your thoughtful suggestions and insightful comments. In response to your valuable feedback, we have diligently revised the manuscript in the following manner.
- I cannot find supplementary files.
Thank you for bringing this issue to our attention. We sincerely apologize for any confusion regarding the supplementary files. To address this concern, we have now attached the files directly to the revision submission form. Please find them there for your convenience. If you encounter any further difficulties or have additional requests, please feel free to let us know.
- Discussion is heavily used to present additional Results. Results should be presented in the Results section.
We appreciate the insightful observation and understand the importance of maintaining clarity in the presentation of results. In response to your suggestion, we have now moved the following information from the Discussion to the Results section.
“As shown by the Masson's trichrome-stained images, fibrosis mainly developed in the myocardium in the DMD group, whereas fibrosis was observed in non-myocardial cells (e.g., around blood vessels) in the non-DMD group.” (Page 2, lines 91–94)
“Additionally, the expression levels of COX1 and COX2, which are located upstream of HPGDS and play a role in the arachidonic acid cascade, were also increased. However, this increase did not reach statistical significance (Figure S2). Similarly, comparing protein expression levels, we found that the expressions of membrane-bound prostaglandin E synthase (mPGES1), COX1, COX2, tryptases, and CD31, were not significantly different among the two groups (Figure S4).” (Page 3, lines 109–114).
- Too many instances of (unpublished data). If the data is unpublished, it is good time to publish that data. If it is not possible, related claims need to be withdrawn.
Thank you for highlighting this concern. We have carefully revised the manuscript to reduce the reliance on the expression "unpublished data." Specifically, we adjusted the wording to align with information reported in existing papers. However, certain results are currently under submission and anticipated to be published in the near future (“We also found that the expressions of serum cardiac troponin I and periostin increased due to cardiac hypertrophy in T3-treated mdx mice were ameliorated by DP1 and DP2 antagonists (unpublished data).” Page 6; lines 196–198). We appreciate your understanding and consideration on this matter.
Based on your precise advice, we have revised the following statements in the Discussion section.
- “However, in our previous study, Dp1 and Dp2 mRNA levels were found to be significantly upregulated in the hearts of mdx mice treated with thyroid hormone (T3) to induce dilated cardiomyopathy [18].” (Page 6, lines 194–196)
“In contrast, Cox1 and Cox2 mRNAs have been found to be upregulated in the hearts of T3-treated mdx mice [18].” (Page 6, lines 226–227)

Round 2
Reviewer 2 Report
Comments and Suggestions for Authors
I believe authors adequately addressed the raised concerns.